# Clinical Characteristics of the End-of-Life Phase in Children with Life-Limiting Diseases: Retrospective Study from a Single Center for Pediatric Palliative Care [note 1]

**DOI:** 10.3390/children8060523

**Published:** 2021-06-19

**Authors:** Fanni Baumann, Steven Hebert, Wolfgang Rascher, Joachim Woelfle, Chara Gravou-Apostolatou

**Affiliations:** Department of Pediatrics and Adolescent Medicine, Friedrich-Alexander-University of Erlangen-Nuremberg, 91054 Erlangen, Germany; fanni.baumann@uk-erlangen.de (F.B.); steven.hebert@uk-erlangen.de (S.H.); wolfgang.rascher@uk-erlangen.de (W.R.); chara.gravou-apostolatou@uk-erlangen.de (C.G.-A.)

**Keywords:** pediatric, palliative care, end-of-life, symptom management

## Abstract

Background: Data on the end-of-life phase of children receiving palliative care are limited. The purpose of this study is to investigate the spectrum of symptoms of terminally ill children, adolescents, and young adults, depending on their underlying disease. Methods: Findings are based on a 4.5-year retrospective study of 89 children who received palliative care before they died, investigating the symptomatology of the last two weeks before death. Results: In this study, the most common clinical symptomatology present in children undergoing end-of-life care includes pain, shortness of breath, anxiety, nausea, and constipation. Out of 89 patients included in this study, 47% suffered from an oncological disease. Oncological patients had a significantly higher symptom burden at the end of life (*p* < 0.05) compared to other groups, and the intensity of symptoms increased as the underlying disease progressed. The likelihood of experiencing pain and nausea/vomiting was also significantly higher in oncological patients (*p* = 0.016). Conclusions: We found that the underlying disease is associated with marked differences in the respective leading clinical symptom. Therefore, related to these differences, symptom management has to be adjusted according to the underlying disease, since the underlying disorder seems to exert an influence on the severity of symptoms and thereby on the modality and choice of treatment. This study is intended to aid underlying disease-specific symptom management at the end-of-life care for children, adolescents, and young adults, with a specific focus on end-of-life care in a home environment.

## 1. Background

Specialized outpatient palliative care teams have been implemented in Germany since legislation related to palliative care was passed in 2007. There are currently 361 such teams in place in the country [1], 35 of which specialize in children and adolescents (specialized outpatient pediatric palliative care—German abbreviation: SAPPV) [2].

However, there are significant differences regarding the focus and setting of specific pediatric palliative care teams even within Germany, with pronounced differences in particular regarding the spectrum of life-limiting disease and whether a more hospital- or home-care oriented palliative care approach is aimed for.

Furthermore, it is necessary to distinguish adult palliative care from pediatric palliative care teams, as there are relevant differences between the two, especially with regard to the underlying disease, length of care, and medication used [3]. Currently, around 50,000 children, adolescents, and young adults live in Germany with a life-threatening (LTC) or life-limiting (LLC) condition. The term life-limiting illness or condition (LLC) is defined as a condition where premature death is usual. Life-threatening illnesses or conditions (LTC) are characterized by a high probability of premature death due to severe illness, although there might be a small chance of long-term survival.

Although the prevalence of LLC in childhood has shown an increase [4], a precise assessment remains difficult. With the increase in the number of patients with LLC, there is a growing need for SAPPV. Initial studies demonstrated that specialized care improves quality of life and significantly reduces suffering, especially in the end of life phase [5]. SAPPV teams enable affected children and young people to be cared for in their home environment, not only in the final stage of their lives, but also within crisis situations, which in certain circumstances can result in temporary stabilization.

Home care solutions at the end of life are a significant desire for many families. Within pediatric palliative care, this is clearly an enormous challenge for both the affected families as well as for caregivers. The stressful symptoms and the needs at the end of life vary not only based on the underlying disease, but also according to age and the degree of development of pediatric patients. There is still too few data on the subject of symptoms and end-of-life suffering in seriously ill children. The aim of this study is to examine whether symptom assessment needs to be stratified according to a patients’ clinical phenotype, including the underlying disease. The primary goal is to characterize associations between symptoms and the clinical condition, including the underlying disease, in order to better anticipate associated complications, thereby potentially enabling earlier and improved symptom control in SAPPV care.

## 2. Methods

Study design and sample: This retrospective analysis covers a time period of 4.5 years. Within the period of 1 January 2016 to 30 June 2020, 89 children who were continuously under the care of the pediatric palliative care department from start of palliative care until their death were included into this study. Patients that were initially under the care of our SAPPV team but subsequently moved elsewhere were not included. The vast majority of these children received both in-patient as well as home care from the SAPPV teams.

Data collection: “PalliCare” is an actual clinical database in which data from our patients are collected. The data included in this registry (“PalliCare”) were used for the present analysis. The following data extracted from PalliCare were collected in an anonymized manner for subsequent analysis: personal data of patients (age, sex, religion), main diagnosis, the burden of symptoms at the beginning of palliative care and during the terminal phase, medication at the beginning of palliative care and during the terminal phase, and place and date of death. All data were documented in a standardized manner.

The terminal phase was defined as the last two weeks before death. Three children died in the first two weeks of life, so in these cases, the evaluation covered their entire lives. For this analysis, underlying diseases were categorized according to two aspects. The first division was made based on clinical aspects, resulting in six groups (congenital genetic or chromosomal diseases, gastroenterological, cardiological, neurological, oncological, and metabolic diseases). However, due to the nature of many palliative conditions, a clinical division does not always allow for a strict allocation. For example, one patient with trisomy 18 suffered from a large ventricular septal defect and therefore exhibited cardiological as well as neurological symptoms due to underlying brain malformation. Due to these sometimes-ambiguous allocations, a second classification was made according to IMPaCCT. The International Meeting for Palliative Care in Children (IMPaCCT) was a group of healthcare professionals from Europe, Canada, Lebanon, and the USA who met in Trento, Italy in 2006 and developed recommendations for defining and identifying standards of care for children with life-limiting illness. IMPaCCT refers to the categorization of children who should receive palliative care. Based on this classification, patients were divided into four groups: Group 1—life-threatening conditions for which curative treatment may be feasible but can fail, where access to palliative care services may be necessary alongside attempts at curative treatment and/or if treatment fails; Group 2—conditions where premature death is inevitable, where there may be long periods of intensive treatment aimed at prolonging life and allowing participation in normal activities; Group 3—progressive conditions without curative treatment options, where treatment is exclusively palliative and may commonly extend over many years; and Group 4—irreversible but non-progressive conditions with complex healthcare needs leading to complications and the likelihood of premature death. [6]. The following symptoms were examined in the course of care and evaluated depending on the illness: pain, shortness of breath, restlessness/fear, nausea/vomiting, and constipation.

### Statistical Analysis

All data were evaluated and analyzed with the SPSS statistics software program. The results of the statistics are given as total number (*n*) and percentage (%) as well as median, range, and standard deviation (SD). The Chi-square test was used to compare categorical variables and the Mann–Whitney U test was used for continuous variables. A *p*-value of < 0.05 was considered significant.

## 3. Results

### 3.1. Patient Characteristics

The characteristics of the patients examined are summarized in Table 1. The median length of care before death was 102 days (range 3–807 days). The median age of our patients at death was six years (range 4 days–36 years). Adult patients were included partly because of developmental delay and partly because of a malignant disease diagnosed in childhood. As already mentioned, patients were divided by underlying diseases based on clinical aspects and according to IMPaCCT. Of all the patients, 47.2% (*n* = 42) suffered from an oncological disease, which ultimately led to death. The most common oncological diagnosis was diffuse midline glioma WHO grade IV (*n* = 14), while the most common non-oncological diseases were trisomy 18 (*n* = 2), short bowel syndrome (*n* = 2), cardiomyopathy (*n* = 4), hypoplastic left heart syndrome (*n* = 4), spastic cerebral palsy (*n* = 4), and cystic fibrosis (*n* = 3).

### 3.2. Symptoms at the Beginning of Care and at the End of Life

Oncological patients had a significantly higher symptom burden at the end of life (*p* = 0.042). A higher symptom burden was defined by a higher number of symptoms. Severity of symptoms was not included in the calculation of symptom burden due the heterogeneous age group of our sample, making a common severity grading of symptoms difficult. We systematically analyzed the drug treatment of symptoms. However, other non-medical means were used in addition, but an analysis of non-medical symptom control was not within the scope of this study. There was a clear increase in the intensity of symptoms as the underlying disease progressed. At the beginning of care, 14 (15.7%) children/adolescents experienced states of anxiety and restlessness. In the terminal phase, 51 patients (57.3%) suffered from the same symptom. A total of 11 patients (12.3%) complained of shortness of breath at the beginning of care; this number rose to 49 (55%) over the investigated period of time. The most common causes of shortness of breath were pulmonary metastases, cardiac decompensation, effusions, and infections. In the sample, 22 (24.7%) patients reported pain at the beginning of the SAPPV care. In the terminal phase, more than half (56%) of the patients suffered from pain (*n* = 50). A total of 15 patients (16.8%) experienced nausea and/or vomiting. This number increased to 29 (32.5%) during the course of the study. The prevalence of constipation often seems to be underestimated; in our study, 21% (*n* = 19) of the patients suffered from this symptom at the start of care, a number that also increased over time (40%, *n* = 36).

### 3.3. Symptoms According to Underlying Diseases

An important aspect of the study was to examine the individual symptoms related to the underlying illness (See Figure 1).

Of all the metabolically ill patients, 70% (*n* = 7) experienced fear and restlessness at the end of their life. Cardiological patients suffered mostly from dyspnea (84.6%, *n* = 11), which can be explained primarily by cardiac decompensation and pulmonary edema. Of all the metabolically ill patients, 80% (*n* = 8) complained of shortness of breath in the terminal phase. Pain occurred significantly more frequently in oncological patients (*p* = <0.01), in total in 86% (*n* = 36) (see Figure 1a). A total of six oncological patients (14% of all oncological patients) had mild/no pain symptoms during basic treatment with dexamethasone and infrequent non-opioid analgesics at the end of their life. All six children suffered from a brain tumor. Of the oncological patients, 45% (*n* = 19) complained of nausea/vomiting, partly due to therapy, partly because of increased intracranial pressure in the context of a brain tumor. Oncological patients vomited significantly more often than non-oncological patients (*p* = 0.016) but vomiting also occurred in 38% (*n* = 5) of all cardiac patients. Of the metabolically ill patients, 60% of (*n* = 6) complained of constipation, as well as 47% (*n* = 20) of all oncological patients and 43% (*n* = 7) of all neurological patients. Of the patients who experienced constipation, 86% (*n* = 31) were under treatment with opioids (for pain or shortness of breath), suggesting a putative causal role of opioid therapy.

### 3.4. Medication in the Terminal Phase

The median value of drugs (which were used to control symptoms) per patient was three (range 0–8). Drugs that were not taken as part of symptom control were not included in this analysis. Most of the drugs were given to oncological patients (3.5 per patient). The most frequently used drugs in our patients are summarized in Table 2.

A total of 59 patients (66.2%) were treated with strong opioids, specifically morphine (*n* = 41), as well as hydromorphone (*n* = 16) and fentanyl (*n* = 2). In comparison, only seven (7.8%) patients received weak opioids in the terminal phase of their lives. A total of 46 patients (51.6%) were treated with non-opioid analgesics, the most common drugs being metamizole, ibuprofen, and paracetamol. Patients with neuropathic pain (*n* = 8) were treated with pregabalin or gabapentin (9%).

Restlessness and anxiety were frequently treated with benzodiazepines (35 patients, 39%), mostly with lorazepam and/or midazolam. Seven patients (7.8%) received benzodiazepines for convulsive disorders without restlessness or fear. Eight patients (9%) received chloral hydrate as reliever medication.

Therapy with CBD (cannabidiol)/THC (tetrahydrocannabidiol) has evolved as an option to support ongoing analgesic treatment when other therapy options have already been exhausted and patients did not respond sufficiently to conventional therapy. A total of six patients (6.7%) received CBD/THC supplements.

A total of 25 patients (28%) received corticosteroids at the end of their life as part of symptom control. In this study, indications for therapy with corticosteroids included their use as co-analgesics mainly for neuropathic and bone pain, as antiemetics, and as treatment for cerebral edema.

Regarding nausea and vomiting, eight patients (9%) were treated with dimenhydrinate and 11 patients (12%) with ondansetron. Four patients (4.5%) received neuroleptics (levomepromazine). With regard to the treatment of constipation, laxatives were only used in 28% of the cases (*n* = 25). In our patient collective, dyspnea was mainly treated with a combination of opioids and benzodiazepines (44%).

### 3.5. Types of Application

Almost 56% (*n* = 48) of patients received their medication mainly orally. In cases where oral administration was not possible and/or a safe central venous access was available, therapy was administered intravenously (36%, *n* = 31). In addition, subcutaneous routes of application increasingly play a role in pediatric palliative medicine; 7% (*n* = 6) of our patients received their analgesics and sedatives via subcutaneous routes. Only 1.2% (*n* = 1) of the patients received transdermal analgesic therapy. Laxatives were applied orally or rectally.

### 3.6. Place of Death

Of our patients, 58% (*n* = 52) died at home in the presence of their family. A further 26% (*n* = 23) died as in-patients of a pediatric ward and only 16% (*n* = 14) spent their last days on an intensive care unit. Most of the children who died at home received their medication orally (69%, *n* = 36), while 17% (*n* = 9) received their medication intravenously (mainly analgesics and sedatives). Only children who died at home received drugs for symptom control via subcutaneous application (a total of 17.5% of the patients who died at home, *n* = 9). A total of 59% of the patients (*n* = 22) who spent the last phase of their lives in a hospital received their medication intravenously.

## 4. Discussion

Our study demonstrates that at the end-of-life phase of children receiving palliative care, the most common distressing symptoms were pain, dyspnea, vomiting, anxiety, restlessness, and constipation. Pain is one of the most common symptoms at the end of life, experienced by 56% of patients in our study. This number is also consistent with other literature, where pain was reported from 45 to 80% [3,7,8,9]. In our sample, 55% of patients suffered from dyspnea in the terminal phase; this number varies between 30–80% in the corresponding literature [3,7,8]. Vomiting is reported at a frequency of 23–63% [3,7,9]; in our study, 32.5% suffered from nausea/vomiting at the end of life. With regard to the frequency of constipation and anxiety/restlessness in these patients, there are still few data available. There is a clear connection between the stressful symptoms at the end of life and the underlying disease. Studies that only examined oncological patients reported a higher symptom burden regarding pain [7,9], whereas children with non-malignant diseases showed a higher number of other symptoms (e.g., epileptic seizures, spasticity, etc.) [3].

Distressing symptoms at the end of life frequently pose a great challenge: children often suffer from pain which, frequently, is difficult to control [10]. Appropriate drug treatment for stressful symptoms is best achieved individually, depending on the patient’s illness and his/her medication history. With regard to pediatrics, there are only a few standardized recommendations for drug-based symptom control at the end of life. To a varying degree, the same drugs seem to be used worldwide [11,12]. In this retrospective analysis, children were given an average of three drugs to control symptoms. This seems to be a small number compared to those described in other studies where patients died in in-patient care settings [7,12], showing that adequate symptom control does not seem to be dependent on the number of medications given. In our cohort, all other medications, with few exceptions (e.g., anticonvulsants, antiarrhythmics), were discontinued at the beginning of palliative treatment. In our sample, 74% of patients received opioids (highly potent and weak opioids) at the end of life. This seems to be relatively high in comparison to other studies [3]. The different distribution of underlying diseases can serve as an explanation, since almost 50% of our patients suffered from a malignant disease, which according to our study, is associated with a significantly higher pain burden. It should not be neglected that pain perception is still underestimated in neuropediatric patients with a developmental disorder, therefore these patients often receive less analgesia.

An increasing number of children with LLC/LTC are dying at home [13]. Between 2009 and 2014, 61% of our patients died at home [14], and a similar tendency is visible for the period between 2016 and June 2019 (58%). In the corresponding literature, this number is also reported around 50% [7,9,15]. With the help of SAPPV, dying under adequate symptom control outside a hospital setting seems feasible. Intensive medical, nursing, and psychosocial support can be provided, but not all children with LLC have the opportunity to die at home. The situation of witnessing, accompanying, and supporting the death of one’s own child at home is often overwhelming. In addition, parents often have to be able to provide nursing and medical tasks (medication, wound care, recognize stressful symptoms, etc.), which is also not a possibility for each family. A correlation between the education of the parents and the place of death has been reported [16]; further important factors include religion and culture [17]. Furthermore, time is decisive with regard to the place of death. A slowly progressive course (e.g., in case of oncological diseases) enables parents to prepare for the terminal phase. Forward-looking planning related to symptom control, a possible emergency situation, and ultimately the death of the patient are all important aspects in palliative medicine. In addition, adequate forward-looking planning regarding the place of death can also enable death in a home environment [18]. In addition to cultural aspects and time, other limiting factors regarding the place of death are important, such as uncertainties in prognosis, discrepancies in therapy goals between family and healthcare professionals, as well as language barriers [19]. Outpatient palliative care not only makes it possible for these children to die at home, but families more often perceive death as peaceful [20]. In addition, it is important to mention that parents whose children died in a hospital seem to regret this decision more frequently [8,9]. It should be noted, however, that these aspects reflect the experience in our palliative outpatient care and may not be generalized in different settings.

Today, palliative medicine is still frequently identified with cancer, but other underlying diseases are increasingly important in childhood. In pediatric palliative care, 63–75% of children suffer from a non-oncological disease [3,21]. A recent Belgian study also showed a similar distribution of underlying diseases, with 50% oncological and 50% non-oncological patients. Interestingly, other studies showed a higher number of non-oncological patients, where only 22% of children had a malignant disease [22].

Every child with a chronic, complex, and life-threatening disease is entitled to palliative care, but there are few standards or uniform recommendations for setting indications [20]. The care of non-oncological patients with complex, life-limiting, sometimes rare diseases often leads to uncertainty among the treating staff. Parents of children with non-oncological diseases receive less support [23]. Distressing symptoms are often not recognized or might be underestimated, but the majority of non-oncological patients suffer from pain at the end of life [24]. Musculoskeletal pain is common, due to contractures, spasticity, or joint dislocations, while in the case of metabolic diseases, pain in the organs due to pronounced enlargement can occur. In addition, due to restricted verbal communication, the treatment of patients with a developmental disorder can become more difficult, which, as previously mentioned, often leads to unsatisfactory symptom control.

In our study, around 11% of the children suffered from a metabolic disease, while literature puts this number between 15–18% [3,25]. Neurological, respiratory, and gastrointestinal symptoms, which can be difficult to treat, were observed in this patient group [25]. About 15% of our patients had a congenital or acquired heart disease. This seems low considering that mortality from congenital heart disease remains high [26]. The explanation here is seen in the fact that children with heart disease more often die as in-patients in intensive care units, e.g., from multiple organ failure under partly invasive intensive care measures [27], and thus do not receive specialized palliative care.

Neonates are a group who are considered special patients in pediatric palliative medicine. There are few data on symptom control at the end of life in newborns, but they mainly suffer from pain, dyspnea, restlessness, and increased secretion production at the end of life [28].

Oncological diseases often require more intensive palliative care but for a shorter period of time [24]. In the case of non-oncological LLCs, due to the often longer care provided, there is a possibility to plan ahead which allows parents to prepare for the last phase of life. This retrospective analysis addresses the issues of children with life-limiting diseases, dying at home in outpatient palliative care.

## 5. Limitations

Limitations of this study include the retrospective nature and variability of documentation at the end of life, as well as the relatively small sample size. Due to the relatively small sample size of 89 patients, the findings of this study cannot be considered nationally representative. In addition, these data do not show the perspective at the end of life from the patient’s point of view but are based on an external assessment from the point of view of parents and staff. Five patients died in external hospitals without information about the treatment and symptoms in their last few days, therefore only data on the period before the in-patient stay could be recorded here.

## 6. Conclusions

Our retrospective analysis underlines that caring for terminally ill children in the home environment is possible and often desirable. Palliative care in the end-of-life phase of children should not only focus on oncological patients; non-oncological life-limiting diseases seem to play an increasingly important role in palliative care. The most common distressing symptoms at the end of life differ depending on the underlying disease of the patient. Regardless of the underlying disease, main symptoms include pain, nausea/vomiting, constipation, shortness of breath, and fear, which can be adequately addressed even in a home-care-centered palliative care setting.

## Figures and Tables

**Figure 1 children-08-00523-f001:**
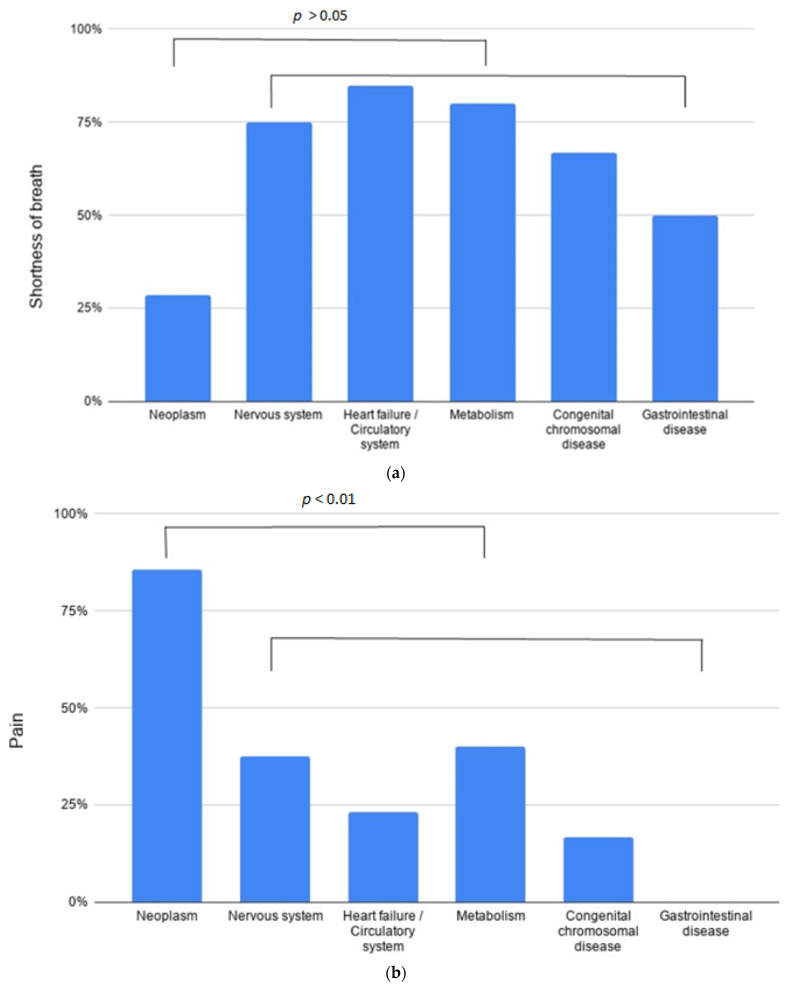
Frequency of symptoms (%) related to the underlying disorder. Figure legend: The first (lower) bracket comprises the group of non-oncological diseases (columns 2–6). These five diseases were compared to the neoplasm group (column 1). The second (upper) bracket indicates the significance of the difference regarding a given symptom (shortness of breath, pain, nausea/vomiting) between patients with oncological and non-oncological diseases. (**a**) Frequency of shortness of breath (%) related to the underlying disorder. We found no significant differences between oncological and non-oncological conditions. (**b**) Frequency of pain (%) related to the underlying disorder. A significant higher number of oncological patients suffered from pain compared to other conditions. (**c**) Frequency of nausea/vomiting (%) related to the underlying disorder. A significantly higher number of oncological patients suffered from nausea/vomiting compared to other conditions.

**Table 1 children-08-00523-t001:** Patient characteristics.

*N* = 89		*N*	%
Gender			
	Male	45	51
	Female	44	49
Age			
(median 6 years)	<1 year	17	19
	1–6 years	28	31
	7–14 years	25	28
	➢ 14 years	19	21
Duration of palliative care			
(median 3.5 months)	<30 days	10	11
	30 d–6 months	52	58
	6 months–1 year	15	17
	➢ 1 year	12	13
Underlying disease (IMPaCCT)			
	Group 1	31	35
	Group 2	15	17
	Group 3	35	39
	Group 4	8	9
Underlying disease (clinical)			
	Congenital chromosomal disease	6	7
	Gastrointestinal disease	2	2
	Heart failure/circulatory system	13	15
	Nervous system	16	18
	Neoplasm	42	47
	Metabolism	10	11
Place of death			
	Hospital-ICU	14	16
	Hospital—except ICU	23	26
	At home	52	58

**Table 2 children-08-00523-t002:** Medication in the terminal phase.

Group of Drugs	Number of Patients (%)
Strong opioids	59 (66.3)
Weak opioids	7 (7.9)
Non-opioid analgetics	46 (51.7)
Benzodiazepines	42 (47.2)
Laxatives	25 (28.1)
CBD/THC	6 (6.7)
Corticosteroids	25 (28.1)
Antidepressants (including amitriptyline)	2 (2.2)
Antihistamine (H1 Antagonist)	8 (9)
5-HT3 receptor antagonist	11 (12.4)
Classical antipsychotics	4 (4.5)
Anticonvulsants (pregabalin, gabapentin)	8 (9)
Chloral hydrate	8 (9.0)

## Data Availability

The datasets used and/or analyzed during the current study are available from the corresponding author on reasonable request.

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
