# Peer review of "Clinical Characteristics of the End-of-Life Phase in Children with Life-Limiting Diseases: Retrospective Study from a Single Center for Pediatric Palliative Care"

_children, 2021, doi:10.3390/children8060523_

Round 1

Reviewer 1 Report

This study investigated the symptomatology of the last two weeks before death among children who received palliative care before death. This study found that olcological disease are frequent among the study population, and oncological patients showed higher symptom of burden before death, and intensity of symptom increased as the underlying disease progressed. The topic of this study is interesting. This study helps to understand better for characteristics of seriously ill children with pediatric palliative care. 

In method section, author should show how the number of final study sample size is 89. Please explain what IMPACCT is.

please check figure legend for figure 1.c. (I think the legend is missing)

This study shows small sample size. I wonder whether the sample is nationally representative or not. The author should consider this point as a limitation. 

Reviewer 2 Report

Thank you for the opportunity to review this manuscript. The manuscript describes the burden of symptoms of children, adolescents and young adults with life limiting diseases and aims to compare the symptoms according to the underlying condition.

While the idea of this study is surely interesting and would give an insight for clinicans to draw attention to certain symptoms depending on the diagnosis, the manuscript itself shows numerous shortcomings and the manuscript needs to undergo a major revision.

1. Wording

The manuscript needs a lanuguage revision. Sentences like "This form of care", "subject of symptoms", "data was" (data = plural), "regarding the quality of the underlying disease" and other examples are improper.

This sentence: "In addition to these 89 children, patients who were no longer in the care of the SAPPV team at the time of death were not included in this analysis" is misleading. "In addition" implies that more patients have been included, but obvioulsy this sentence should express the opposite.

2. Abstract

"Therefore, related to these differences, symptom management has to be adjusted according to the underlying disease."

I don´t think that this is the essence of the study. Symptom management follows symptom severity but not the cause of the symptom.

3. Background:

- LLC and LTC need to be defined in the text. What is the difference? Alternatively, the authors can delte LTC, because the rest of the manuscript mentions only LLC.

- "The primary goal is to characterize these symptoms sufficiently and thus to ensure a satisfactory symptom control in SAPPV care." How does the characterization of the symptoms leads to better symptom control?

4. Methods

- It remains unclear, if the study setting includes only in-patients or home care patients as well?

- It remains unclear wether the „PalliCare“ is a tool for collection oft he data from another source (this is how it reads) or if „PalliCare“ is the actual clinical database that was analyzed by the authors.

- The authors write: „The following data was collected in an anonymized manner in an Excel database“ without mentioning the data. I understand that the „underlying disease“ (meaning main diagnosis?) was categorized. But what about the symptoms? Where did the authors recorded the burden of symptoms? From the handwritten files or from „PalliCare“. And if so, have the symptoms been documented in a standardized way?

5. Results

- What does IMPACCT mean? It is not defined in the text nor in the table. Groups 1,2,3,4 stand for ?

- This sentence belongs to "methods" not to results: „The following symptoms were examined in the course of care and evaluated depending on the illness: pain, shortness of breath, restlessness/fear, nausea/vomiting and constipation.“

- How was a higher symptom burden defined? By calculating the different type of symptoms or severity of symptoms (e.g. pain scale?)

- „All symptoms had to be treated with medication“ I don´t understand this sentence. There has been not a single symptom treated by other means than medication among all patients?

- „of the oncological patients complained of nausea/vomiting, partly due to therapy, partly because of increased intracranial pressure in the context of a brain tumor“ How was this distinguished?

- „86% (n=31) of the patients with constipation were treated with opioids.“ Very misleading. I assume that the patients did not receive opioids to treat constipation and I rather assume that the authors want to express a possible correlation of opioids and side-effects?

- Figure 1a-c: What is the meaning of the brackets in the figure? Why would the authors group the first 4 and the last 5 disease categories?

- There is a discrepancy in the wording: „strong opioids“ vs. Highly potent opoids“ Please use only one term throughout the text.

- „In recent years, an increasing number of children and adolescents seem to have been treated with cannabidiol (CBD) or tetrahydrocannabidiol (THC) in palliative medicine. The main indications are spasticity, nausea, neuropathic pain and anxiety. Therapy with CBD/THC has evolved as an option to support ongoing analgesic treatment, when other therapy options had already been exhausted and patients did not respond sufficiently enough to conventional therapy.“ This does not belong to the "results" paragraph.

6. Discussion

- The second paragraph of the discussion describes the use of medication. I am not sure if this is the aim of this study. The background/methodology describes the aim as "the understanding of symptom burden in the last 2 weeks". But in this paragraph it is mentioned that „In our patient cohort, all other medications, with few exceptions (e.g. anticonvulsants, antiarrhythmics), were discontinued at the beginning of palliative treatment.“ First of all, this is not relevant for the aim of the study questions, secondly it remains unclear how this was assessed? The methodology describes the documentation of symptoms in the last 2 weeks. The discussion part should only discuss what has been measured and what has been ment to be measured.

- „An important aspect is the response to therapy, where treating physicians often have to rely on the assessment of parents. However, parents, in comparison to medical professionals, assess symptoms and perceive the efficacy of therapy differently (7).“ This is interesting but without any context to the aforementioned or following paragraph.

General comment: The study is relevant for palliative care clinicans. However, the manuscript has many shortcomings and I would strongly suggest that a senior colleague should supervisor the manuscript development more carefully. The description of the methodology has significant shortcomings, some parts of the result paragraph discuss the findings already. Abbreveations remain unexplained. The discussion part should focus on the results/findings and discuss around the differences in symptoms depending on the underlying diagnosis. A guiding question for the discussion could be (for example) "does sympom assessment needs to be different in patients with according to the main diagnosis". "Is the same approach of care applicable to all patients regardless of the underlying condition?" These are the relevant questions that follows naturally out of the author´s intention to compare symptoms.

A major revision is needed.

Reviewer 3 Report

Interesting paper, not quite orginal but nevertheless useful to have literature to support what we have known, and i commend the authors for this. The subgroups were grouped in broad terms and it would have interesting to know the diagnoses rather that group patient very broadly. Eg nervous group, these may be children with respiratory, muscular weakness, or scoliosis which impacts on symtpomes and how these may have been managed. Please check the grammer, in some cases reported in present tense others past. This is a retrospective chart review and findings are what was found and not what is currently present.

Round 2

Reviewer 2 Report

Dear authors,

thank you for the revised version of your manuscript. All questions have been answered and ambiguity has been removed.

I have no objection for publication.